# Numina-Lean-Agent: An Open and General Agentic Reasoning System for Formal Mathematics

**Junqi Liu** [* 1 2]  **Zihao Zhou** [* 3 4 2]  **Zekai Zhu** [* 5 2]  **Marco Dos Santos** [6 2]  **Weikun He** [1]  **Jiawei Liu** [2]  **Yunzhou Xie** [7 2]  **Junqiao Zhao** [5]  **Qiufeng Wang** [4]  **Lihong Zhi** [1]  **Jia Li** [2]  **Wenda Li** [8]

## Abstract

Agentic systems have recently become the dominant paradigm for formal theorem proving, achieving strong performance by coordinating multiple models and tools. However, existing approaches often rely on task-specific pipelines and models specialized in theorem proving through large-scale training, limiting flexibility. In this paper, we propose the paradigm that directly uses a general coding agent as a formal mathematics reasoner. This paradigm is motivated by three considerations: (1) a coding agent provides a natural interface for diverse reasoning tasks beyond proving; (2) performance can be improved by swapping the underlying base model without additional training; and (3) Model Context Protocols (MCP) enables flexible extension and autonomous invocation of specialized tools, avoiding complex manual pipeline design. Based on this paradigm, we introduce **Numina-Lean-Agent**, which combines Claude Code with Numina-Lean-MCP to enable autonomous interaction with Lean, retrieval of relevant theorems, informal proving, and auxiliary reasoning tools. Numina-Lean-Agent solves all problems from the William Lowell Putnam 2025 competition (12/12), matching the best closed-source systems. Beyond benchmark evaluation, we further demonstrate its generality by interacting with mathematicians to formalize a recent harmonic analysis paper.

---

*Equal contribution  [1]Academy of Mathematics and Systems Science, University of Chinese Academy of Sciences  [2]Project Numina  [3]University of Liverpool  [4]Xi'an Jiaotong-Liverpool University  [5]Tongji University  [6]University of Cambridge  [7]Imperial College London  [8]University of Edinburgh. Correspondence to: Jia Li <jia@projectnumina.ai>, Wenda Li <wenda.li@ed.ac.uk>.

*Proceedings of the $43^{rd}$ International Conference on Machine Learning*, Seoul, South Korea. PMLR 306, 2026. Copyright 2026 by the author(s).

## 1. Introduction

Formal theorem proving aims to construct machine-verifiable proofs for mathematical theorems within rigorously defined logical systems, such as Lean (2015) and Isabelle (Paulson, 1994). Unlike informal mathematical reasoning, formal verification systems provide tools for automatically and soundly verifying the correctness of proofs. Consequently, these systems establish the foundation for the development of reliable reasoning. Previous advances in neural theorem proving have focused on developing single-model formal provers. The early provers relied on tactic prediction combined with explicit search methods, such as Monte Carlo tree search, to explore the proof space (Lample et al., 2022; Hubert et al., 2025; Xin et al., 2024b). To mitigate the efficiency limitations of search-based methods, subsequent work explored whole proof generation to directly produce complete proofs (Jiang et al., 2022; First et al., 2023). Subsequently, other efforts incorporated informal reasoning to guide tactic generation and proof construction (Wang et al., 2025; Ren et al., 2025; Lin et al., 2025b). Despite notable progress, effectively capturing long-horizon structured reasoning within formal systems remains a central challenge.

More recently, several systems—such as Hilbert (Varambally et al., 2025), AxiomProver (Axiom Math Team, 2025), Seed Prover (Chen et al., 2025b;a), Aristotle (Achim et al., 2025), and Ax-Prover (Breen et al., 2025)—have moved beyond single-model formal provers by introducing agentic workflows that enable provers to interact with formal theorem proving environments and other models. Despite their strong performance, existing agentic proving systems exhibit several limitations: (1) They rely on task-specific reasoning pipelines that are explicitly designed and often coupled with extensively trained formal provers. (2) Most of the systems are closed-source and provide limited implementation details, making it difficult for the larger community to reproduce and extend their work.

In this paper, we propose a paradigm for building a formal mathematics reasoner based on a general coding agent, motivated by three considerations: (1) coding agents provide a native interface for diverse proof-engineering tasks beyond

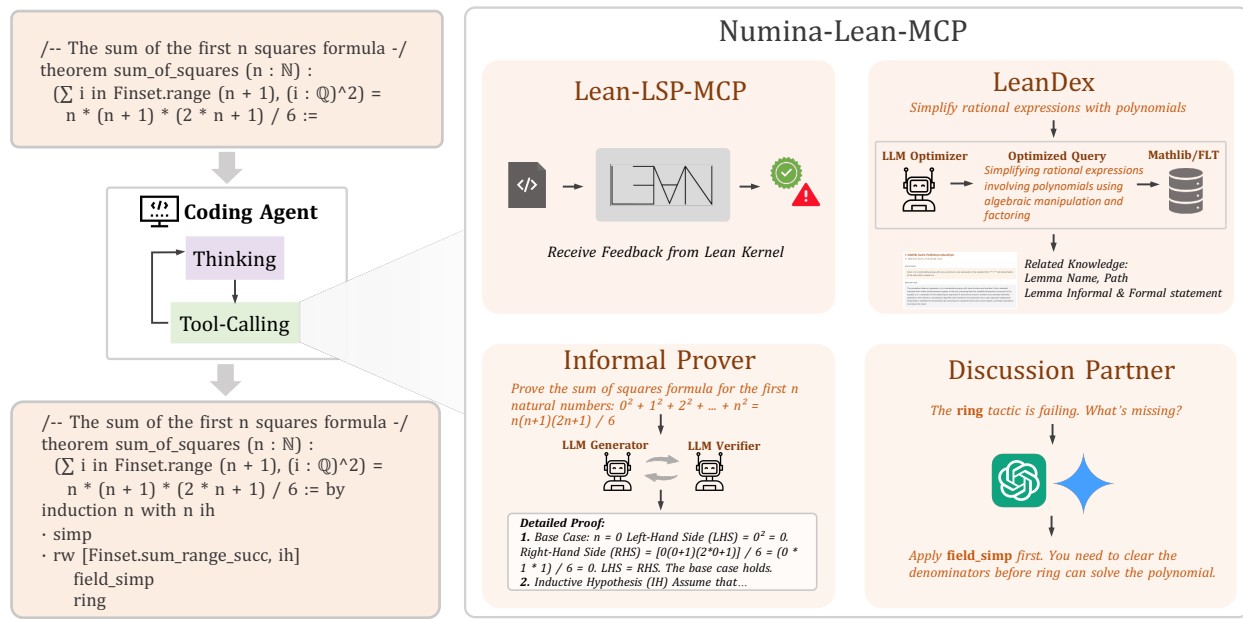

*Figure 1.* Overview of Numina-Lean-Agent, an agentic formal reasoning framework built on Claude Code and Numina-Lean-MCP. The agent autonomously selects and invokes specialized reasoning tools to handle diverse queries.

proving; (2) the underlying base model can be swapped to improve reasoning capability without additional training; and (3) integrating MCP enables plug-and-play extension of specialized reasoning tools that the agent can autonomously invoke based on the query.

Following this paradigm, we propose Numina-Lean-Agent, which couples Claude Code with *Numina-Lean-MCP* to expose a suite of reasoning tools. Numina-Lean-MCP integrates multiple components, including Lean-LSP-MCP (Dressler, 2025) for interacting with Lean, *LeanDex* for semantic retrieval over Lean libraries (e.g., Mathlib), an *Informal Prover* for generating natural language proof, and a *Discussion Partner* module that queries external language models. Together, these modules form Numina-Lean-Agent, a general-purpose formal mathematical reasoning system.

Unlike existing approaches such as Ax-Prover, which relies on a rigid multi-agent pipeline with fixed prompts and pre-determined workflows (e.g., cycling through Orchestrator, Prover, Verifier), Numina-Lean-Agent is built on a general coding agent paradigm. Our model autonomously determines the proof workflow based on current state, rather than following a predefined loop. This flexibility enables seamless scalability—new tools (e.g., via MCP) can be added without redesigning the framework—making our system more robust and general-purpose for formal mathematics.

Using Claude Opus 4.5 (Anthropic, 2025) as its base model, Numina-Lean-Agent successfully solved all 12 problems from the William Lowell Putnam 2025 competition, achiev-ing state-of-the-art performance. This result matches the closed-source system AxiomProver (Axiom Math Team, 2025) and surpasses Harmonic's Aristotle (Achim et al., 2025) and Seed-Prover 1.5 (Chen et al., 2025a). We report all solutions along with their computational costs and proof lengths in Section 4. Notably, on certain problems such as Problem B1, Numina-Lean-Agent produced more concise proofs than AxiomProver (Axiom Math Team, 2025) and Seed-Prover 1.5. Beyond standard automated proving, Numina-Lean-Agent serves as a general mathematical reasoning system, enabling mathematicians to engage in interactive "vibe proving". We demonstrate this paradigm by working with human experts to formalize a recent harmonic analysis paper about an upper bound of a multiplicative constant in the Brascamp–Lieb inequalities (Bénard & He, 2025), with the details of the interactive process reported in Section 5.

## 2. Related Work

Automated theorem proving remains a difficult problem in artificial intelligence (Polu et al., 2022). Recent efforts have combined Lean 4 with large language models, enabling systems to reach near–IMO-level mathematical reasoning performance (Hubert et al., 2025). In general, LLM-based provers can be divided into two main paradigms: step-wise interaction (Xin et al., 2025; 2024a;b) and whole-proof generation (Wang et al., 2025; Ren et al., 2025; Lin et al., 2025a;b).

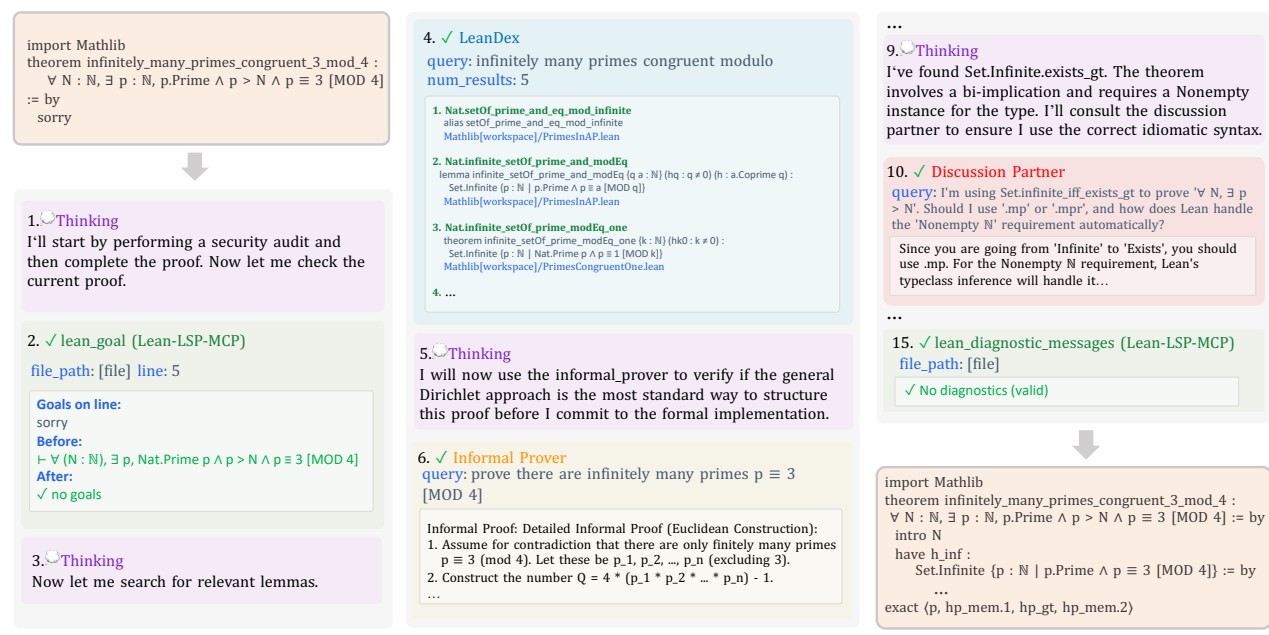

*Figure 2.* Numina-Lean-Agent dynamically leverages reasoning tools to autonomously navigate and advance formal proofs

As the capabilities of general models improve, more and more research is focusing on designing agentic workflows to better utilize the capabilities of existing models. HILBERT (Varambally et al., 2025) proposes an agentic framework that combines informal reasoners with formal provers to guide proof construction. Ax-Prover (Breen et al., 2025) proposes an agentic workflow that connects general-purpose LLMs to interactive proof assistants via the MCP to equip general-purpose LLMs with Lean tools from the lean-lsp-mcp repository (Dressler, 2025). Seed-Prover 1.5 (Chen et al., 2025a) trains a formal prover through large-scale agentic reinforcement learning, emphasizing repeated interaction with the Lean compiler and related tools. In addition, Ax-iomProver (Axiom Math Team, 2025), developed by Axiom Math, adopts an autonomous multi-agent ensemble architecture and has achieved a perfect score on Putnam 2025. These systems highlight the growing effectiveness of agentic proving systems.

## 3. Numina-Lean-Agent

### 3.1. Overview

As shown in Figure 1, Numina-Lean-Agent is an agentic formal theorem proving framework built on Claude Code and Numina-Lean-MCP. Functioning as an autonomous agent, it can dynamically select and invoke the appropriate reasoning tools within Numina-Lean-MCP to handle diverse queries and complete complex formal reasoning tasks.

This framework brings three key advantages: (1) a language

model serves as a natural and general interface for diverse queries. This significantly broadens the applicability of the system, enabling it to handle not only proving tasks, but also auxiliary procedures such as proof golfing and statement repair. (2) modern coding agents already possess strong formal reasoning abilities, which further improve as the base model advances. Instead of training a specialized prover from scratch, we can directly inherit and continuously benefit from improvements in base models. (3) MCP-based extensibility makes it easy to integrate new tools for reasoning. The MCP plugin mechanism allows the system to be easily extended with tools that assist reasoning. Crucially, the language model can autonomously decide when to invoke these tools, rather than relying on manually designed workflows.

### 3.2. Numina-Lean-MCP

To enhance the formal reasoning capability of Numina-Lean-Agent, we integrate a set of tools into Numina-Lean-MCP. These tools fall into four categories, each improving the system along a distinct and complementary dimension: (1) Interaction with the Lean kernel. We integrate Lean-LSP-MCP, which enables the agent to interact with the Lean kernel environment. (2) Knowledge retrieval. Retrieving relevant knowledge from Mathlib is crucial for constructing correct and efficient proofs. To support this, we introduce LeanDex, which maintains up-to-date library access and generates clearer search intents. (3) Informal proving. We incorporate an informal prover to generate informal solutions before formalization. Previous work has shown that

an informal solution helps models with planning and tactic generation (Wang et al., 2025). (4) Multi-model deliberation. We introduce a discussion partner that enables multiple models to participate in the reasoning process. This cooperation improves the quality and robustness of reasoning. Together, these tools improve the agent's reasoning ability from the perspectives of environment interaction, knowledge retrieval, informal proving, and multi-model deliberation. Figure 2 illustrates how the agent invokes tools to advance the proof. We describe each tool in detail below.

**Lean-LSP-MCP.** Lean-LSP-MCP (Dressler, 2025) is an MCP server explicitly designed for the Lean theorem prover. Acting as a bridge between LLMs and the Lean kernel via the Language Server Protocol (LSP), this server enables agents to access diagnostics, goal states, hover documentation, while also integrating Loogle – the official Lean search tool. Consequently, it enhances the agents' capabilities in retrieving relevant theorems, understanding context, and performing automated reasoning within Lean projects

**LeanDex.** We introduce LeanDex, a theorem-search tool for Lean that enables semantic retrieval under Lean v4.26.0. Existing tools have notable limitations: Loogle requires queries to follow a strict syntactic format, while local search mechanisms are largely confined to searching within a local project. LeanExplore, on which LeanDex builds, was developed for Lean 4.19 and is no longer compatible with the latest Lean ecosystem due to limited maintenance. Moreover, LeanExplore only supports Mathlib, which is increasingly insufficient as the Lean community develops and maintains additional domain-specific libraries such as FLT and PhysLean. We rebuild the underlying database using up-to-date versions of Mathlib and other repositories, including FLT and PhysLean, and maintain it through a regular monthly update cycle. We also regenerate the informal descriptions of definitions and theorems using DeepSeek-Chat v3.1, improving the consistency and quality of the semantic index. Given a natural-language query, LeanDex first applies an LLM-based query-optimization step to better capture the intended mathematical meaning, and then performs semantic retrieval over Lean objects from multiple libraries. Since the initial retrieval results may contain noise, LeanDex further applies an LLM-based re-ranking stage to prioritize the most relevant definitions and theorems. In this way, LeanDex provides an agentic semantic theorem-search interface that is more flexible than purely syntactic search tools and broader than local project search.

**Informal Prover.** Inspired by recent IMO-level agents for informal proving (Huang & Yang, 2025), we implement an Informal Prover that generates natural language proofs. The system comprises two models: a Generator and a Verifier. The Generator is responsible for generating informal solutions, while the Verifier assesses the correctness of the generated solutions. These two models interact in an iterative refinement loop. When the Verifier identifies errors in the generated proof, it will provide feedback to the Generator. In the next iteration, the Generator refines its solution on the basis of both the previous solution and the Verifier's feedback. This process continues until the Verifier accepts the solution as correct or a maximum number of iterations is reached, which we set to 20.

To improve the reliability of verification, the Verifier evaluates each candidate solution independently three times. A solution is accepted only if all three verification passes judge it to be correct. In our implementation, we use *Gemini-3-Pro-Preview* for both the Generator and the Verifier.

**Discussion Partner.** In scientific research, discussion is widely recognized as an effective cognitive tool. By exchanging diverse viewpoints and reasoning paths, researchers often overcome blind spots and generate new insights. Inspired by this, we designed and implemented `discuss_partner` to assist the formalization process.

Specifically, the tool enables Claude to seek assistance during Lean formalization: when encountering obstacles—such as proof bottlenecks, dilemmas in strategy selection, or ambiguities in intermediate lemmas—the primary model can initiate discussions with other LLMs. These models analyze the current state from complementary perspectives and propose candidate ideas or alternative proof paths. This multi-model collaboration improves exploration efficiency and increases robustness and success rates in formalization.

*Remark* 3.1. All ablation studies in Section 4 are conducted under the setting where the Informal Prover and the discussion partner are jointly enabled and treated as a coupled component.

### 3.3. Subagent System

To improve robustness in long-horizon formalization—where the proof state and context can grow substantially—we leverage Claude Code's built-in subagent capability to enable context isolation and focused proof development. Rather than relying on a manually specified decomposition, the main agent autonomously decides when to spawn a subagent: when a lemma becomes sufficiently complex or repeatedly stalls progress, it can open a fresh, tightly scoped subagent instance to pursue that lemma in isolation under a restricted context budget. Concretely, the agent extracts the target lemma (together with only the minimal required local dependencies) into a temporary file and asks the subagent to complete a standalone Lean proof there, thereby sharply reducing the active context and proof-state clutter. If successful, the verified lemma will be merged back into the main file by the subagent; otherwise, the subagent returns a structured failure report which the main agent can use to guide subsequent attempts and avoid repeating unproductive

*Table 1.* Performance comparison of Numina-Lean-Agent against other methods

| **PUTNAM2025** | A1 | A2 | A3 | A4 | A5 | A6 | B1 | B2 | B3 | B4 | B5 | B6 |
|---|---|---|---|---|---|---|---|---|---|---|---|---|
| ARISTOTLE | ✓ | ✓ | ✓ | ✓ | | ✓ | ✓ | ✓ | ✓ | | ✓ | ✓ |
| SEED-PROVER 1.5 | ✓ | ✓ | ✓ | ✓ | | ✓ | ✓ | ✓ | ✓ | ✓ | ✓ | ✓ |
| AXIOM | ✓ | ✓ | ✓ | ✓ | ✓ | ✓ | ✓ | ✓ | ✓ | ✓ | ✓ | ✓ |
| NUMINA-LEAN-AGENT | ✓ | ✓ | ✓ | ✓ | ✓ | ✓ | ✓ | ✓ | ✓ | ✓ | ✓ | ✓ |

*Table 2.* Time spent comparison of Numina-Lean-Agent against other methods (Unit: minutes)

| **PUTNAM2025** | A1 | A2 | A3 | A4 | A5 | A6 | B1 | B2 | B3 | B4 | B5 | B6 |
|---|---|---|---|---|---|---|---|---|---|---|---|---|
| ARISTOTLE | 30 | 60 | **30** | 180 | – | **60** | 150 | **25** | 40 | – | 420 | **180** |
| SEED-PROVER 1.5 | 60 | **30** | 120 | 240 | – | 240 | 540 | 360 | **30** | 120 | 240 | **180** |
| AXIOM | 110 | 180 | 165 | **107** | 518 | 259 | 270 | 65 | 43 | **112** | 254 | 494 |
| NUMINA-LEAN-AGENT | **27** | 81 | **30** | 169 | 2040 | 89 | **55** | 136 | **30** | 308 | **88** | 797 |

*Table 3.* Code length comparison of Numina-Lean-Agent against other methods

| **PUTNAM2025** | A1 | A2 | A3 | A4 | A5 | A6 | B1 | B2 | B3 | B4 | B5 | B6 |
|---|---|---|---|---|---|---|---|---|---|---|---|---|
| ARISTOTLE | 45 | 195 | 103 | 291 | – | 123 | 223 | 108 | 70 | – | 291 | 280 |
| SEED-PROVER 1.5 | 631 | 469 | 927 | 1095 | – | 881 | 849 | 1613 | 584 | **628** | 2499 | 2594 |
| AXIOM | 556 | 458 | 1089 | 825 | **1878** | **468** | 1179 | **346** | 302 | 993 | 1310 | **862** |
| NUMINA-LEAN-AGENT | **365** | **401** | **422** | **605** | 3263 | 835 | **328** | 690 | **292** | 648 | **929** | 1820 |

trajectories. In our experiments, subagent invocations are executed strictly sequentially and in a blocking fashion (i.e., the main agent waits for each subagent to complete before proceeding).

## 4. Evaluation

### 4.1. Performance

We evaluated Numina-Lean-Agent on the Putnam 2025 benchmark and compared its performance with other existing provers. We show a performance comparison with other systems in Table 1. In particular, we used the formal statements provided by Seed-Prover 1.5. Moreover, all operations in our system were strictly sequential (i.e., no parallel execution), and Internet search was disabled for all API calls to prevent solution retrieval via online search. Under these settings, Numina-Lean-Agent achieved state-of-the-art performance, successfully solving all problems (12 / 12) from Putnam 2025. We used an approximate budget of $50 per problem by default; due to substantially higher difficulty and longer proof search trajectories, we allocated a larger budget of approximately $1000 for A5 and approximately $300 for B6. These values are intended to reflect relative computational effort rather than exact accounting.

As shown in Table 2, we report a comparison of per-problem solving time between Numina-Lean-Agent and other representative agents on Putnam 2025. Despite the fact that Numina-Lean-Agent operates without any parallel execution, it demonstrates notable efficiency advantages on a subset of problems, achieving shorter solving times than competing methods on several instances.

In Table 3, we further compare the proof length generated by different provers. For fairness, we remove all comments and blank lines from the final Lean code and report line counts. Compared to AxiomProver and Seed-Prover 1.5, Numina-Lean-Agent produces shorter proofs on many problems, with particularly large gains on A3, B1, and B5. We note that step-based provers have an inherent advantage in producing very short proofs, and therefore the proofs generated by Numina-Lean-Agent are generally longer than those produced by Aristotle. Nevertheless, when compared with other agentic provers under a similar setting, Numina-Lean-Agent consistently yields more concise formalizations on most problems, demonstrating its effectiveness in generating compact and efficient formal proofs.

### 4.2. Putnam 2025 A5

Putnam 2025 A5 is a representative case where enabling the subagent system is beneficial. In addition to its conceptual difficulty, A5 exposes a practical gap in available Mathlib infrastructure: several supporting facts are not directly reusable off-the-shelf, so the formalization must develop a

nontrivial amount of lower-level material before the main argument can be assembled. As a result, the final Lean proof is comparatively long. Without the subagent system, the model often fails to maintain sufficient focus across this long proof trajectory. In our A5 run, the agent invokes subagents to sustain attention on bottleneck lemmas while continuing the surrounding formalization.

### 4.3. Ablation Study

#### 4.3.1. GENERALIZATION ACROSS BASE MODELS

To evaluate the generalization capability of our framework, we conduct a model-replacement study in which the base model of Claude Code is varied while keeping the overall agent architecture unchanged. Nevertheless, the proposed framework itself is not inherently tied to Claude Code as a specific code agent. In principle, Claude Code can be replaced by alternative code-agent tools such as Gemini CLI or Kimi CLI, with compatibility achieved through corresponding adaptations of the MCP interface. This interchangeability further reflects the system-level generalization of our framework.

*Table 4.* Base model ablation results on Putnam 2025 problems.

| PROBLEM | CLAUDE OPUS 4.5 | DEEPSEEK V3.2 | KIMI K2 |
|---------|------------------|----------------|---------|
| A1 | ✓ | ✓ | |
| A2 | ✓ | ✓ | |
| A3 | ✓ | | |
| B1 | ✓ | | |
| B2 | ✓ | | |
| B3 | ✓ | ✓ | |

Building on our previous results, we further evaluate the framework using DeepSeek V3.2 and Kimi K2 Thinking as base models. The results in Table 4 indicate that our framework exhibits a certain degree of generalization across these models. However, both models show notable limitations in instruction following and token efficiency, preventing them from matching the performance achieved with Claude Opus 4.5. Specifically, under the same problem settings, DeepSeek typically requires more than an order of magnitude (10×) larger token budget to solve comparable problems, revealing a substantial disadvantage in token efficiency. Moreover, both models demonstrate weaker instruction-following ability than Claude Opus 4.5: as the reasoning process extends and the context grows longer, DeepSeek often fails to reliably adhere to previously specified prompts, while Kimi is more prone to completely forgetting earlier instructions during long-horizon interactions.

*Table 5.* Ablation of the coding agent on a six-problem subset of Putnam 2025. The base model is fixed to Claude Opus 4.5 for all configurations.

| Coding agent | A1 | A2 | A3 | B1 | B2 | B3 |
|--------------|----|----|----|----|----|----|
| Claude Code | ✓ | ✓ | ✓ | ✓ | ✓ | ✓ |
| OpenCode | ✓ | ✓ | ✓ | | ✓ | ✓ |

*Table 6.* Tool ablation results on a subset of six Putnam 2025 problems. We report end-to-end solving time in minutes. "–" indicates that the problem was not solved within the allocated budget.

| PROBLEM | FULL | W/O LEANDEX | W/O LEANDEX + IF PROVER + DISCUSS |
|---------|------|-------------|-----------------------------------|
| A1 | 27 | 51 | 46 |
| A2 | 81 | 185 | 57 |
| A3 | 30 | 30 | – |
| B1 | 55 | – | – |
| B2 | 136 | 136 | – |
| B3 | 30 | 97 | 87 |

#### 4.3.2. ABLATION OF THE CODING AGENT

To evaluate the dependence of our framework on the coding-agent implementation, we conduct an additional ablation study on a six-problem subset of Putnam 2025. In this experiment, we fix the base model to Claude Opus 4.5 and compare two different coding-agent: Claude Code and OpenCode. The results are shown in Table 5. Due to the high computational cost of each full proving run, this experiment is conducted on a representative subset rather than the full benchmark. Nevertheless, the results provide evidence that the performance of our framework is not tied to a single coding-agent implementation.

With Claude Code as the coding agent, the system solves all six problems in the subset. When Claude Code is replaced by OpenCode, a fully open-source coding agent, while keeping the same base model, the system still solves five out of six problems, failing only on B1. This suggests that our method does not fundamentally rely on the opaque implementation details of Claude Code. Instead, the framework can be instantiated with alternative coding agents, provided that they support the necessary interaction pattern with the Lean environment and external tools.

Overall, these results indicate that our approach is coding-agent-agnostic to a meaningful extent. The remaining performance gap between Claude Code and OpenCode may arise from differences in tool-use stability, long-horizon planning, and robustness to iterative Lean feedback.

### 4.3.3. TOOL ABLATION STUDY

We conduct partial ablation experiments on six Putnam 2025 problems, as reported in Table 6. Due to the large number of possible tool combinations and the high computational cost of each proving run, we focus on ablating several key components: LeanDex, the Informal Prover, and the Discussion Partner. The results show that each component contributes meaningfully to the overall performance of the system.

First, enabling LeanDex substantially improves end-to-end solving efficiency. For problems A1, A2, and B3, the full configuration achieves significantly shorter solving times than the setting without LeanDex, with the solving time nearly halved in some cases. For example, removing LeanDex increases the solving time from 27 to 51 minutes on A1 and from 81 to 185 minutes on A2. Moreover, the system fails to solve B1 when LeanDex is removed. This indicates that semantic retrieval can efficiently provide relevant mathematical knowledge without requiring the model to know the exact Mathlib names of theorems and definitions. Notably, even in the without-LeanDex setting, the system still has access to basic retrieval tools based on exact or pattern matching, such as Loogle. Therefore, the observed performance gap further highlights the advantage of semantic retrieval over purely syntactic retrieval.

Second, the results on A3 and B2 highlight the importance of the Informal Prover and the Discussion Partner for mathematical reasoning and proof planning. When LeanDex, the Informal Prover, and the Discussion Partner are jointly removed, the system fails to solve A3, B1, and B2 within the allocated budget, whereas the full configuration succeeds on these problems. This suggests that these components do not merely improve retrieval efficiency, but also provide high-level solution strategies, informal reasoning, and interactive guidance that are crucial for tackling more complex formalization tasks. In particular, the Informal Prover helps generate structured mathematical plans, while the Discussion Partner supports open-ended exploration and refinement of proof ideas during the solving process.

Finally, the ablation results also suggest that these tools are not uniformly beneficial for every problem. For relatively simple problems such as A1 and A2, the configuration without LeanDex can achieve shorter solving times than some configurations that enable the Informal Prover. One possible explanation is that these problems already fall within the direct reasoning capability of the base model, making additional informal-proof generation redundant or even slightly inefficient. This suggests that an adaptive tool-selection strategy may further improve efficiency by invoking high-level reasoning tools only when the problem requires them.

We also report the number of invocations of each tool across the twelve Putnam problems and give more comprehensive

*Table 7.* Informal Prover ablation results. We report end-to-end solving time in minutes.

| PROBLEM | GEMINI 3 PRO | GPT 5.2 PRO | DEEPSEEK V3.2 |
|---------|-----------|-----------|-----------|
| A1 | 27 | 46 | 92 |
| A2 | 81 | 222 | 145 |
| A3 | 30 | 51 | 239 |
| B1 | 55 | - | - |
| B2 | 136 | 264 | 233 |
| B3 | 30 | 70 | 143 |

analysis in the Appendix D.

### 4.3.4. ABLATION OF INFORMAL PROVER CHOICE

As shown in Table 7, if the main agent and tool configuration is fixed, the choice of the underlying model for the Informal Prover has a substantial impact on end-to-end efficiency and robustness. Overall, Gemini 3 Pro is the most reliable option on this subset: it achieves the best or near-best solving time on A1/A2/A3/B2/B3 (e.g., 81 minutes on A2 and 30 minutes on B3) and it also completes B1 (55 minutes). In contrast, GPT 5.2 Pro is consistently slower on the problems it solves (e.g., 222 minutes on A2 and 264 minutes on B2) and fails to finish B1 within the budget, suggesting that stronger or more verbose informal reasoning does not necessarily translate into faster formalization and can instead introduce additional verification and alignment overhead. Deepseek V3.2 is slower overall (e.g., 239 minutes on A3 and 143 minutes on B3) and also fails on B1, indicating lower token efficiency and weaker ability to produce concise, formalization-friendly proof sketches.

## 5. Formalizing Brascamp Lieb with Numina-Lean-Agent

### 5.1. Blueprint Generation

Formalizing a complex theorem in Lean is a long-horizon task with dense dependencies. When Claude Code is asked to directly prove the final statement, it often gets trapped in local dead ends. We therefore introduce a **blueprint** as an explicit planning layer that decomposes the global goal into a sequence of verifiable subgoals.

A blueprint is a design-document-style artifact consisting of (1) required definitions and notation, (2) a curated list of intermediate lemmas with suitable granularity, and (3) the final theorem whose proof largely composes these lemmas. Dependencies are recorded explicitly (e.g., through `\uses{...}`), forming a DAG that determines the proving order and reduces ambiguity during search.

Importantly, blueprint generation is *recursive* and tightly

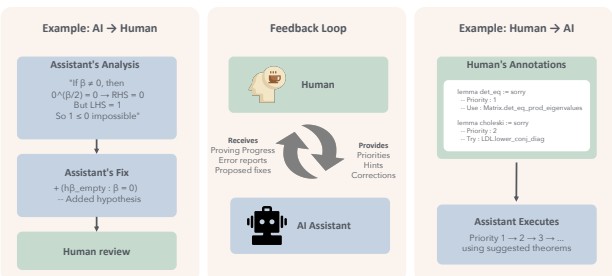

*Figure 3.* The collaborative workflow between human experts and Numina-Lean-Agent in the formalization of Effective Brascamp–Lieb Inequalities.

coupled to the formalization loop rather than a one-shot pre-processing step. As the agent attempts to discharge lemmas in Lean, compilation feedback and proof-state inspection may reveal that an informal step is incorrect, underspecified, or split at an unsuitable granularity. In such cases, the agent revisits and refines the blueprint (e.g., rephrasing statements to match Lean interfaces, or inserting missing intermediate lemmas) and then continues formalization with the updated plan. To improve robustness, the agent can also invoke external discussion models (e.g., Gemini) to propose alternative decompositions or repairs when the current blueprint repeatedly leads to bottlenecks.

Overall, the blueprint plays the role of a high-level mathematical plan: a stronger mathematical reasoner is used to decompose a difficult statement into a sequence of small, checkable steps, while Claude Code focuses on turning these steps into machine-verifiable Lean proofs. Crucially, Lean feedback (missing lemmas, mismatched interfaces, etc.) provides concrete signals that are fed back to revise the blueprint, yielding a closed-loop "plan–formalize–refine" workflow that stabilizes long-horizon formalization.

### 5.2. Human-AI Cooperation

We design a human–machine collaborative interaction framework for Numina-Lean-Agent, enabling human experts to work together with the agent by writing hints and modifying the Blueprint. We conduct a collaborative case study based on a preprint **Effective Brascamp–Lieb Inequalities** (Bénard & He, 2025), published in November 2025. We show the formal statement of the main theorem of the Effective Brascamp–Lieb inequalities in Appendix A.1. In this experiment, a mathematician, a Lean formalization expert, and Numina-Lean-Agent jointly cooperate to formalize the results of the paper. During a period of less than two weeks of intermittent collaboration, the two human experts and the agent completed the formalization of more than **8,000** lines of Lean code. During this process, the agent autonomously introduced approximately **70** new definitions, lemmas, and theorems, illustrating its ability to

actively extend the formal library and participate in large-scale sustained formalization efforts. We present the formal statement of the main theorem in Appendix A.1.

When formalizing more involved arguments, the agent sometimes chose to further decompose the proof, introducing additional intermediate lemmas that were more fine-grained than those in the original blueprint. This behavior appears to be a form of adaptive proof decomposition tailored to the demands of formal verification.

We structure the human–AI interaction as a lightweight closed-loop workflow: humans specify intent and conduct review, while the agent carries out high-frequency exploration and implementation. Formalization is executed in a sequence of runs under fixed token budgets guided by the current blueprint. Each run produces a structured summary of progress and remaining blockers, and its code changes are recorded as a git commit, yielding an auditable trajectory that supports comparison and rollback.

This verification-driven "plan–advance–review" cycle turns long-horizon formalization into reviewable incremental checkpoints, reducing human cognitive load while improving controllability and traceability. Human intervention is intentionally low-friction: we provide occasional, localized guidance via short inline comments near the relevant code (e.g., pointing to Mathlib lemmas, setting priority of statement or suggesting a proof direction). Lean's diagnostic feedback pinpoints errors by line number and supports rapid navigation, allowing the agent to reliably consume nearby hints in context and close the loop through verified progress.

Moreover, compared with other specialized prover models, our agent is not restricted to theorem proving alone, but instead exhibits strong general-purpose reasoning capabilities. For a given formal statement whose correctness is not known in advance, traditional approaches typically can only attempt to prove both the original statement and its negation in parallel. In contrast, during our formalization of the Brascamp–Lieb inequalities, we observed that our agent is able to actively reason about the validity of the statement itself during the proof process. When it detects that a statement is incorrect, it can autonomously revise the statement accordingly. This ability to dynamically inspect and revise the problem formulation during formalization has not been present in previous provers. We present concrete examples of this behavior in the Appendix B.

### 5.3. Limitations

In some cases, the system generates Lean code that is overly verbose or poorly structured. In our experiments, we primarily tasked the agent with two types of "sorry"s of varying difficulty. When a "sorry" required only local reasoning within an already well-structured proof, the agent typically

filled the gap with high-quality code. However, when a "sorry" corresponded to proving an entire lemma, the agent was generally able to solve the goal, but the resulting code was often verbose and less concise than desired. This reveals a key limitation: although the system can handle complex proof goals, the readability and structural quality of the generated formalization degrade for larger or more intricate tasks.

Our system occasionally struggles with type-level issues, which can significantly slow down the proof process. For example, in one case, the agent failed completely—not because of difficulties in the core mathematical argument but because of a type conversion from Real to NNReal. Such type-level constraints are rarely made explicit in informal mathematics, so the agent had difficulty reconstructing the required structure on its own. After reviewing the proof workflow and handling type conversions in advance to make the formalization path more "type-friendly", the agent was able to complete the remaining proof successfully. This case highlights the inherent gap between informal and formal proofs and underscores the challenge that type-level requirements can pose for automated reasoning systems.

Moreover, despite their strong problem-solving capabilities in automated theorem proving, current agents still exhibit a clear gap between functional correctness and formal elegance. Although agent-generated proofs pass Lean's compiler checks, experienced Mathlib contributors often perceive them as overly result-oriented, relying on verbose and low-level tactic scripts. Compared to human-written Mathlib code, these proofs lack structured abstraction and idiomatic use of higher-level patterns, leaving substantial room for improvement in conciseness, readability, and conformity to Mathlib's community standards.

## 6. Conclusion

Our results indicate that a general-purpose coding agent, when augmented with a small set of carefully designed tools, can serve as a competitive and flexible formal mathematical reasoner. The system achieves state-of-the-art performance, matching closed-source alternatives. Compared to task-specific prover pipelines, this approach reduces training and engineering overhead and enables seamless improvements as the underlying base model evolves.

The evaluation highlights several open challenges. First, although the agent can often reach a correct proof, the resulting Lean code is not always at the level of abstraction and readability expected for long-term maintenance in Mathlib-style development. Second, long-horizon proofs remain brittle: tool calls, search decisions, and statement choices can compound, and the system may require explicit decomposition (e.g., subagents and blueprints) to stabilize

progress.

More broadly, our findings also suggest that strong performance on standalone theorem-proving benchmarks does not necessarily translate to superior effectiveness in agent-based settings. In particular, although DeepSeek achieves higher scores than Claude on the Lean4 miniF2F benchmark, this advantage does not persist when the models are used as the base models of our agent framework, where we observe a reversed performance trend. This highlights that agent-based formal reasoning places additional demands on the base model, such as long-horizon instruction following, structured interaction with external tools, and robustness under multi-round feedback.

Looking forward, we view two directions as particularly promising: (i) improving proof-quality objectives (readability, modularity, and reuse) alongside correctness; and (ii) enriching MCP tool ecosystems with domain-specific capabilities while keeping interfaces simple enough for autonomous orchestration. We hope that our open-source project will support the neural theorem proving community and accelerate future research.

## Acknowledgement

Junqi Liu and Lihong Zhi are supported by the National Key R&D Program of China 2023YFA1009401 and Strategic Priority Research Program of Chinese Academy of Sciences under Grant XDA0480501. Wenda Li is funded by the the AI for Math Fund. Thanks to XTX Markets for sponsoring and supporting Project Numina. Zihao Zhou and Qiufeng Wang are supported by the National Natural Science Foundation of China under No.62436009.

## Impact Statement

This work studies the use of general-purpose coding agents for formal mathematical reasoning. The contributions are primarily methodological, focusing on agent design, tool integration, and interaction with formal systems such as proof assistants. The proposed paradigm aims to improve the flexibility and extensibility of automated reasoning systems rather than to target specific real-world deployment scenarios. We do not identify any direct societal risks uniquely introduced by this work.

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

# A. Effective Brascamp–Lieb inequalities

Brascamp–Lieb inequalities are generalizations of a few very commonly used inequalities in mathematical analysis, such as the Cauchy-Schwarz inequality, the Hölder inequality and the Loomis-Whitney inequality. Research around these inequalities has drawn a lot of attention in the past decade in the domain of harmonic analysis, partly because of their connection with Kakeya estimates.

The preprint (2025) on which we have based our collaborative case study is a recent and cutting-edge contribution to this research topic. It is focused on giving viable estimate of a multiplicative constant in the Brascamp–Lieb inequalities.

We have successfully formalized Theorem 1.4 of (2025), the main theorem therein.

**Theorem A.1** (Upper bound). *Let $(\mathcal{D}, \mathbf{R}, T)$ be a localised regularised Brascamp–Lieb datum, let $\alpha \in \mathbb{R}_{>0}^J$ and $\beta \in \mathbb{R}_{\geq 0}$. Assume $\mathcal{D}$ is $(\alpha, \beta)$-perceptive and $\mathrm{rk}_{\alpha_j}(\ell_j) = \dim H_j$ for each $j \in J$. Then, writing $d = \dim H$, we have*

$$\mathrm{BL}(\mathcal{D}, \mathbf{R}, T) \leq d^{\frac{\mathcal{A}(\mathcal{D})}{2}} \mathcal{E}(\mathcal{D}) \prod_{j \in J} \alpha_j^{-q_j \dim H_j} N(\mathcal{D}, \mathbf{R}, T)^{\frac{\mathcal{A}(\mathcal{D})-d+\beta}{2}} \|T^{-1}\|^{\frac{\beta}{2}}. \tag{1}$$

The following is its formal statement.

## A.1. Formal statement of the main theorem.

```
theorem upperBound {J : Type*} [Fintype J]
    {E : Type*} [NormedAddCommGroup E] [InnerProductSpace ℝ E] [FiniteDimensional ℝ E]
    {F : J → Type*} [(j : J) → NormedAddCommGroup (F j)]
    [(j : J) → InnerProductSpace ℝ (F j)] [(j : J) → FiniteDimensional ℝ (F j)]
    (hE : Module.finrank ℝ E ≠ 0)
    (D : locRegDatum E F) (α : J → NNReal) (β : NNReal) (hα : ∀ i, 0 < α i)
    (hP : D.IsMetricPercep α β)
    (hS : ∀ j : J, (D.map j).EssentialRank rfl (α j) = Module.finrank ℝ (F j)) :
    let M_max := (D.loc + Σ j, (D.weight j) · (D.map j).adjoint ∘ₗ (D.reg j) ∘ₗ (D.map j))
    loc_reg_constant_g D ≤
      (Module.finrank ℝ E : NNReal)^(D.Acuity / 2 : ℝ) *
      (Π j, (D.weight j)^(- (D.weight j : ℝ) * Module.finrank ℝ (F j) / 2)) *
      (Π j, (α j)^(- (D.weight j : ℝ) * Module.finrank ℝ (F j))) *
      ‖M_max.toContinuousLinearMap‖₊^((D.Acuity.toReal - Module.finrank ℝ E + β) / 2) *
      ‖(D.loc.equivOfDetNeZero D.pos_loc.2).symm.toContinuousLinearMap‖₊^(β.toReal / 2) :=
    by sorry
```

## B. Self-Correction of Formal Statements during Formalization.

The Lean snippet below illustrates a concrete instance of statement-level self-correction triggered by a degenerate case. The lemma `upperBound_empty_case` isolates the regime `hn : ¬Nonempty n`, in which the ambient space `EuclideanSpace ℝ n` has dimension 0. In this setting, the usual upper-bound expression involves terms such as $\|\cdots\|_+^{(\beta/2)}$ and related exponentiated constants; the proof becomes valid only after adding the auxiliary hypothesis `hβ_empty : β = 0`, which forces these potentially nontrivial factors to collapse to the intended trivial bound. Importantly, this is not merely a Lean technicality: the code makes explicit that the original informal statement was underspecified in the empty-dimension case, and it repairs the claim by strengthening the assumptions exactly where needed.

```
/-- When n is empty (dimension 0), the upper bound holds trivially.
    This lemma handles the degenerate case where the base space has dimension 0.
    In this case, both the LHS and RHS simplify to specific values and the inequality
    holds.
-/
lemma upperBound_empty_case {J : Type*} [Fintype J]
    {n : Type*} [Fintype n] [DecidableEq n]
    {m : J → Type*} [(j : J) → Fintype (m j)]
    (α : J → NNReal) (β : NNReal) (hα : ∀ i, 0 < α i)
    (D : locRegDatum (EuclideanSpace ℝ n) (fun j ↦ EuclideanSpace ℝ (m j)))
    (hP : D.IsMetricPercep α β)
    (hS : ∀ j : J, (D.map j).EssentialRank (α j) = Fintype.card (m j))
    (hn : ¬Nonempty n)
    (hβ_empty : β = 0)   -- Added: when n is empty, β must be 0 for the inequality to hold
    (A : (j : J) → EuclideanSpace ℝ (m j) →ₗ[ℝ] EuclideanSpace ℝ (m j))
    (hA : ∀ j, (A j).IsPosDef ∧ A j ≤ D.reg j) :
    (loc_constant_g_of n m D.1 A (fun j => (hA j).1) : ENNReal) ≤
      ↑((NNReal.rpow (Fintype.card n) (D.Acuity / 2)) *
      Π j, NNReal.rpow (D.weight j) (- D.weight j * Fintype.card (m j) / 2) *
      Π j, NNReal.rpow (α j) (- D.weight j * Fintype.card (m j)) *
      NNReal.rpow ‖(D.loc + Σ j, (D.weight j) · (D.map j).adjoint ∘ₗ (D.reg j) ∘ₗ
        (D.map j)).toContinuousLinearMap‖₊
        ((D.Acuity - Fintype.card n + β) / 2) *
      NNReal.rpow
        ‖(D.loc.equivOfIsUnitDet (by simp
[D.pos_loc.2])).symm.toLinearMap.toContinuousLinearMap‖₊
        (β / 2)) := by
```

## C. Case Study: Comparative Analysis of Putnam 2025 B1 and B5 Proofs

This case study compares the solutions generated for Putnam 2025 problems B1 and B5. On these problems, Numina-Lean-Agent produced proofs faster than Aristotle, Seed-Prover 1.5, and AxiomProver: 55 minutes for B1 (compared to 150, 540, and 270 minutes) and 88 minutes for B5 (compared to 420, 240, and 254 minutes).

Beyond these metrics, the systems exhibit different global strategies. For Problem B1 (geometry), Numina-Lean-Agent avoided coordinate calculations entirely by using connectedness of the circle and the intermediate value property for the continuous distance function. In contrast, AxiomProver relied on constructive analytic geometry, manually defining rotation vectors and trigonometric parameterizations, while Seed-Prover 1.5 relied on algebraic manipulations of vector norms. Aristotle adopted a hybrid approach, mapping the 2D problem into 1D distance intervals.

A similar contrast appears in Problem B5 (number theory). Numina-Lean-Agent transformed the descent condition into a geometric region-counting problem on a modular hyperbola, utilizing involution symmetry. Conversely, AxiomProver reduced the problem to solving quadratic congruences via Legendre symbols. Aristotle converted the count into a global summation problem, while Seed-Prover 1.5 defaulted to a raw arithmetic approach, manually constructing sets and proving elementary modular inverse properties over a thousand lines of code.

Compared to other models, Numina-Lean-Agent distinguishes itself by often adopting high-level structural approaches rather than defaulting to low-level mathematics. It effectively utilizes the Mathlib library to streamline its proofs. Instead of re-implementing standard results, the agent delegates this work to existing high-level theorems like `isPreconnected_sphere` or `Polynomial.card_roots`. Consequently, the resulting code is shorter, and the

*Table 8.* Tool invocation counts across the twelve Putnam problems.

| Tool | A1 | A2 | A3 | A4 | A5 | A6 | B1 | B2 | B3 | B4 | B5 | B6 |
|------|----|----|----|----|----|----|----|----|----|----|----|----|
| `lean_diagnostic_messages` | 44 | 58 | 53 | 131 | 1683 | 103 | 40 | 59 | 69 | 121 | 17 | 212 |
| `gemini_informal_prover` | 1 | 1 | 1 | 0 | 10 | 2 | 1 | 2 | 1 | 13 | 1 | 11 |
| `lean_leandex` | 16 | 42 | 5 | 36 | 180 | 17 | 34 | 40 | 27 | 34 | 10 | 16 |
| `lean_loogle` | 7 | 27 | 14 | 45 | 153 | 27 | 18 | 30 | 13 | 39 | 8 | 41 |
| `lean_local_search` | 14 | 34 | 5 | 8 | 144 | 5 | 21 | 43 | 28 | 12 | 11 | 27 |
| `discussion_partner` | 0 | 2 | 0 | 0 | 212 | 0 | 2 | 0 | 0 | 0 | 1 | 9 |
| `lean_goal` | 10 | 13 | 21 | 47 | 714 | 25 | 7 | 33 | 13 | 45 | 10 | 101 |
| `lean_hover_info` | 6 | 11 | 12 | 4 | 85 | 6 | 13 | 19 | 2 | 4 | 0 | 23 |
| `lean_run_code` | 2 | 29 | 1 | 112 | 5 | 1 | 59 | 23 | 31 | 21 | 125 | 0 |
| `lean_declaration_file` | 1 | 3 | 0 | 1 | 1 | 0 | 2 | 3 | 1 | 1 | 0 | 0 |
| `lean_completions` | 3 | 0 | 0 | 1 | 7 | 1 | 0 | 1 | 0 | 2 | 0 | 0 |
| `lean_multi_attempt` | 1 | 0 | 0 | 5 | 35 | 0 | 0 | 1 | 0 | 2 | 0 | 76 |
| `lean_build` | 2 | 0 | 1 | 0 | 11 | 2 | 0 | 1 | 1 | 0 | 0 | 0 |

high-level approach and readability make the proofs significantly more interpretable for humans, utilizing semantically named lemmas to produce a natural flow that mimics mathematical writing.

## D. Tool ablation analysis

Table 8 reports the number of invocations of each tool across the twelve Putnam problems. The usage pattern illustrates the complementary roles of different components in the agent. For most problems, the agent primarily relies on Lean-facing tools, including diagnostic messages, goal inspection, theorem search, and code execution. These tools support the standard interaction loop with Lean: writing proof code, checking compiler feedback, inspecting the resulting proof state, and modifying the proof accordingly.

It is not hard to find that the discussion partner is used much selectively. Its usage becomes particularly prominent on the most difficult problems, especially A5. On A5, the Gemini informal prover is invoked only 10 times, whereas the discussion partner is invoked 212 times. This large gap suggests that the discussion partner is not merely a substitute for the informal prover. Rather, it serves a distinct role: it enables open-ended mathematical dialogue, helping the agent clarify intermediate arguments, explore alternative proof strategies, and repair local reasoning gaps that are difficult to resolve through a single-shot informal proof.

This distinction is further supported by our ablation study on two challenging problems, A5 and B6. When the discussion partner is removed, both problems become unsolved. This indicates that the discussion partner plays a critical role in problems that require sustained informal reasoning in addition to local Lean-level feedback.

More broadly, the tool invocation statistics reveal a layered structure in the agent's problem-solving process. The first layer consists of high-frequency Lean interaction tools, especially `lean_diagnostic_messages` and `lean_goal`. These tools provide compiler feedback and proof-state information, forming the basic write–verify–modify loop that sustains the agent's interaction with Lean. The second layer consists of retrieval tools, including `lean_loogle`, `lean_leandex`, and `lean_local_search`. These tools are used during the regular proof-search process to locate relevant definitions, lemmas, and theorems from Mathlib and other Lean libraries. The third layer consists of external reasoning tools, such as `gemini_informal_prover` and `discussion_partner`. Unlike the first two layers, these tools are not invoked uniformly across all problems; instead, they are activated when the agent encounters difficult mathematical bottlenecks that require high-level reasoning, alternative proof ideas, or global strategic discussion.

Overall, this layered usage pattern suggests that the agent's performance does not depend on any single tool in isolation. Rather, it emerges from the interaction between low-level Lean verification, theorem retrieval, and high-level informal mathematical reasoning.

