# OpenReview forum: "Numina-Lean-Agent: An Open and General Agentic Reasoning System for Formal Mathematics"
_ICML.cc/2026/Conference — ICML 2026 regular_

### Official Review · Reviewer_c9nC · 2026-03-10

**Soundness:** 3
**Presentation:** 3
**Significance:** 4
**Originality:** 2
**Overall Recommendation:** 4
**Confidence:** 5

**Summary:**

The study introduces a Lean harness including MCPs for coding agents, specifically Claude Code. This is motivated by existing agentic theorem provers relying on task-specific pipelines and specialized prover models, which the authors claim limits flexibility. They propose using a general-purpose coding agents as a mathematics reasoner as these models show strong performance on reasoning tasks and are extendable through MCPs and swapping the underlying base models. This decouples the agentic provers from specific tools and models. The proposed harness includes Lean-LSP-MCP (Lean language server), LeanDex (semantic retrieval), an Informal Prover Model and a Discussion Partner Model. The experiments measure theorem proving performance on the Putnam 2025 benchmark and show that the proposed Numina Lean Agent achieves strong proving performance on par or better than closed-source theorem proving agents. The analysis also includes time and and generated token evaluations. The authors show that the framework can be extended by later introducing “blueprint” generation.

**Compliance With Llm Reviewing Policy:**

Affirmed.

**Final Justification:**

I believe that my original evaluation for the paper is still appropriate. The work has relevant practical applications and a narrow scientific contribution. The authors presented additional information and results that provide support to some of the unsupported claims in the manuscript. Yet, the current experimental setup and results rely on the generalizability of closed-source coding agents.
I like the additional ablation study on the tool-call statistics and would have liked a thorough discussion and interpretation, i.e. including success rates or contributions towards the goal per tool, in the  camera-ready version of the manuscript.

**Key Questions For Authors:**

- Do you have experimental results of an agentic provers showing limited extensibility to new tools or domains?
- line 105: missing reference [Seed-Prover 1.5](https://arxiv.org/abs/2512.17260)
- Please elaborate on the novelty of the Numina-Lean-Agent Key in contrast to existing approaches, such as Ax-Prover?
- Can you quantify the trade-off of optimizing a models theorem proving specific tool-usage capabilities vs. its capability to generalize to other tools?
- What are the usage statistics over the different tools provided to the agent?
- Do you have an ablation study on the performance contributions for each tool?

**Limitations:**

yes

**Strengths And Weaknesses:**

## Strengths

- The experiments show strong performance results on Putnam 2025. These results are relevant as they match performance of frontier models and products, which are not openly describing their methodology.
- The experiments include ablations on code length and time per problem, with the Numina-Lean-Agent generating shorter lean proofs and requiring less time per problem.
- Include an ablation study for the informal prover models.
- Show that the system design allows extendability to new components, i.e. interaction with humans and introducing blueprints.

---

## Weaknesses

- The authors highlight the opaqueness of closed-source systems, however the proposed system also relies on opaque closed-source models, specifically Claude and Gemini.
- Argue that prover models trained to use specific tools are not extendable and cannot handle auxiliary tasks (proof golfing and statement repair), but do not provide any experimental evidence to back up the claims.
- The proposed agent is heavily relying on Claude-4.5 Opus. The ablations studies in Table 4 show weak transfer to deepseek-v3.2 and kimi k2. Authors acknowledge this and reference weaker instruction-following capabilities and long-context reasoning capabilities.
- The included ablation studies do not evaluate all proposed tools in isolation. Furthermore, no breakdown of the tool usage is provided.
- The proposed methodology has limited novelty, but its application to theorem proving is relevant.

---

> ### Author Rebuttal · Authors · 2026-03-31
>
> Thank you for your review!
>
> ## Weekness
>
> **The authors highlight the opaqueness of closed-source systems, however the proposed system also relies on opaque closed-source models, specifically Claude and Gemini.**
>
> Our contribution is an open-source framework, not a specific model. Our ablation experiments show that Claude and Gemini can be replaced with open-source alternatives (e.g., DeepSeek) with some performance trade-off, demonstrating the framework's flexibility across different model choices.
>
> **Prover models trained to use specific tools are not extendable and cannot handle auxiliary tasks (proof golfing and statement repair)**
>
> We conducted qualitative evaluations on Kimina-Prover-Preview, DeepSeek-Prover-V2, and Goedel-Prover-V2, finding that: (1) Kimina and DeepSeek are limited to single-pass generation and cannot perform multi-turn proof refinement; (2) Goedel-Prover-V2, while supporting error-based repair, lacks zero-shot adaptability—introducing auxiliary requirements like proof golfing led to instruction-following failures or output degradation. We will clarify these observations in the revision.
>
> ## Question
>
> **Do you have experimental results of an agentic prover showing limited extensibility to new tools or domains?**
>
> Existing agent provers like Seed Prover are closed-source (precluding direct comparison) and rely on trained tool-use capabilities—extending to new tools requires data collection and retraining. Open-source alternatives like Hilbert adopt fixed pipelines without general tool-use interfaces, making tool extension costly and requiring substantial pipeline modifications. In contrast, our MCP-based design decouples tools from the model: adding a new tool simply requires implementing a lightweight MCP server, with no retraining or pipeline changes needed.
>
> **Missing reference Seed-Prover 1.5**
>
> We referenced it on line 96.
>
> **Please elaborate on the novelty of the Numina-Lean-Agent Key in contrast to existing approaches, such as Ax-Prover?**
>
> Unlike existing approaches such as Ax-Prover, which relies on a rigid multi-agent pipeline with fixed prompts and predetermined workflows (e.g., cycling through Orchestrator, Prover, Verifier), Numina-Lean-Agent is built on a general coding agent paradigm. Our model autonomously determines the proof workflow based on current state, rather than following a predefined loop. This flexibility enables seamless scalability—new tools (e.g., via MCP) can be added without redesigning the framework—making our system more robust and general-purpose for formal mathematics.
>
> **Can you quantify the trade-off of optimizing a models theorem proving specific tool-usage capabilities vs. its capability to generalize to other tools?**
>
> We apologize that we lack sufficient resources for a rigorous quantitative comparison. However, our general tool-using approach offers two key advantages over tool-specific fine-tuning: (1) Extensibility: it can readily incorporate new tools without bias toward a fixed training-time toolset; (2) Task generalization: it naturally adapts to tasks beyond proving (e.g., proof golfing, statement repair), whereas specialized provers may degrade outside their narrow scope.
>
> **What are the usage statistics over the different tools provided to the agent?**
>
>  Below are the statistics of tool invocation counts across all 12 problems from Putnam 2025:
> |Tool|a1|a2|a3|a4|a5|a6|b1|b2|b3|b4|b5|b6|
> |---|---|---|---|---|---|---|---|---|---|---|---|---|
> |lean_diagnostic_messages|44|58|53|131|1683|103|40|59|69|121|17|212|
> |gemini_informal_prover|1|1|1|0|10|2|1|2|1|13|1|11|
> |lean_leandex|16|42|5|36|180|17|34|40|27|34|10|16|
> |lean_loogle|7|27|14|45|153|27|18|30|13|39|8|41|
> |lean_local_search|14|34|5|8|144|5|21|43|28|12|11|27|
> |discussion_partner|0|2|0|0|212|0|2|0|0|0|1|9|
> |lean_goal|10|13|21|47|714|25|7|33|13|45|10|101|
> |lean_hover_info|6|11|12|4|85|6|13|19|2|4|0|23|
> |lean_run_code|2|29|1|112|5|1|59|23|31|21|125|0|
> |lean_declaration_file|1|3|0|1|1|0|2|3|1|1|0|0|
> |lean_completions|3|0|0|1|7|1|0|1|0|2|0|0|
> |lean_multi_attempt|1|0|0|5|35|0|0|1|0|2|0|76|
> |lean_build|2|0|1|0|11|2|0|1|1|0|0|0|
>
> **Do you have an ablation study on the performance contributions for each tool?**
>
> We conducted partial ablation experiments in Table 5 on six Putnam 2025 problems. Due to the large number of possible tool combinations and the high computational cost of each proving run, we focus on ablating several key tools. The results demonstrate their significant impact: removing LeanDex substantially increases solving time (e.g., A1: 27→51 min, A2: 81→185 min) and leads to failure on B1. Further removing LeanDex together with the informal prover and discussion partner results in additional failures (A3, B1, and B2), suggesting that each tool contributes meaningfully to the overall system performance.

---

> > ### Author Rebuttal · Reviewer_c9nC · 2026-04-01
> >
> > Thank you for the detailed response. The authors addressed my concerns and provided an ablation on the tool calls. There are concerns that were not addressed in the rebuttal: there are not experimental results on the claim that tool-use in theorem proving models is limited, requiring the use of coding agents. The argument that performance for other coding agents (apart from Claude) in the proposed framework is decreased does not support that claim.
> > I like the additional insight into the effectiveness of the different tools and would have liked to see a more detailed analysis in the manuscript. This would also support the authors premise of using general-purpose agents that can externalize mathematical problem solving to the required expert tools.

---

> > > ### Author Response · Authors · 2026-04-02
> > >
> > > We sincerely thank you for your valuable suggestions. We apologize if our original phrasing was ambiguous and caused any misunderstanding. We would like to clarify our claims and address your comments in detail.
> > >
> > > **1. Clarification on "limiting extensibility to new tools or domains" in the second paragraph of the introduction**
> > > - For systems relying on fixed pipelines (such as Hilbert), every reasoning step is pre-defined. **Integrating a new tool into such a system is not plug-and-play**; it requires substantial engineering effort to refactor the codebase and redesign the entire pipeline logic.
> > > - On the other hand, models like Seed-Prover-1.5 achieve three tools integration ( Lean verification, Mathlib search, and Python execution) through post-training. Consequently, extending such systems to support a completely new tool typically necessitates **collecting new tool-specific data and undergoing further fine-tuning to acquire the specific tool-calling capabilities.**
> > >
> > > **2.We conducted an additional experiment to help to prove that tool-use in theorem proving models is limited**
> > >
> > > We provided a new search tool LeanDex to the Goedel-Prover-v2-32B and Kimina-Prover 72B, without any post-training for this tool.
> > >
> > > Here is the prompt:
> > > ````
> > > Complete the following Lean 4 code:
> > > ```lean4
> > > {formal_statement}
> > > ```
> > >
> > > Before producing the Lean 4 code to formally prove the given theorem, provide a detailed proof plan outlining the main proof steps and strategies.
> > > The plan should highlight key ideas, intermediate lemmas, and proof structures that will guide the construction of the final formal proof.
> > > You are given access to an external tool called LeanDex, which allows you to search for relevant lemmas and theorems in Mathlib. Use this tool whenever you want to search for helpful lemma.
> > > ````
> > >
> > > and query message：
> > > ```python
> > > TOOLS = [
> > >     {
> > >         "type": "function",
> > >         "function": {
> > >             "name": "LeanDex",
> > >             "description": (
> > >                 "Search for relevant lemmas and theorems in Mathlib. "
> > >                 "Use natural language, Lean identifiers, or proof states as the query."
> > >             ),
> > >             "parameters": {
> > >                 "type": "object",
> > >                 "properties": {
> > >                     "query": {
> > >                         "type": "string",
> > >                         "description": "Search query, e.g. 'Cauchy Schwarz', 'List.sum', or a proof state",
> > >                     },
> > >                     "num_results": {
> > >                         "type": "integer",
> > >                         "description": "Max number of results to return",
> > >                         "default": 5,
> > >                     },
> > >                 },
> > >                 "required": ["query"],
> > >             },
> > >         },
> > >     }
> > > ]
> > >
> > > response = client.chat.completions.create(
> > >     model="Goedel-LM/Goedel-Prover-V2-32B",
> > >     messages=messages,
> > >     tools=TOOLS,
> > >     temperature=0.7,
> > >     max_tokens=16384,
> > > )
> > > ```
> > >
> > > We selected a problem from Putnam 2025 and conducted a Pass@32 test on both the Goedel and Kimina models. The results showed that **neither model generated any tool calls in their output**. Further observation revealed that the models didn't even show any intention to invoke Leandex. Furthermore, when we directly input a query asking Goedel to search for "Chinese Remainder Theorem," the model got caught in a hallucination of proving the theorem itself, consistently failing to trigger a tool call to retrieve the relevant theorem.
> > >
> > > **3. Regarding the detailed analysis of the effectiveness of the different tools**
> > >
> > > We completely agree with your insightful comment. A more comprehensive analysis of how each tool contributes to the performance indeed strengthens our core claim.
> > >
> > > Based on the core dimension of call frequency, we can divide the various tools into three progressively different ecological niches.
> > >
> > > - At the bottom are the extremely frequently called `lean_diagnostic_messages` and `lean_goal`, which are used to read compiler information and extract the current proof goal, forming the most basic "write-verify-modify" loop to maintain the Agent's operation.
> > >
> > > - Up the next layer are retrieval tools, such as `lean_loogle`, `lean_leandex`, and `lean_local_search`. During progress, they are used to retrieve existing theorems from Lean's mathematical library, which is part of the regular problem-solving process.
> > >
> > > - At the very top are the important tools for interacting with external models. This includes `discussion_partner` and `informal_prover`. These tools are usually infrequent resources, mainly used to handle extremely difficult tasks (such as the A5 and B6). When the Agent gets stuck, it uses them to introduce other perspectives on reasoning logic and to engage in global strategy discussions with other models.
> > >
> > > In our later version, we will significantly expand the corresponding section to include more comprehensive and detailed analysis.
> > >
> > > If you have any further questions, or want us to conduct further experiments, we are happy to answer.

---

### Official Review · Reviewer_T9xY · 2026-03-13

**Soundness:** 2
**Presentation:** 2
**Significance:** 4
**Originality:** 3
**Overall Recommendation:** 4
**Confidence:** 4

**Summary:**

This paper introduces the Numina-Lean-Agent, an automated theorem-proving framework that leverages the general-purpose programming assistant Claude Code integrated with domain-specific MCP tools for Lean.

The authors develop Numina-Lean-MCP, a modular system that integrates existing community tools (such as Lean-LSP-MCP and informal provers) with two novel components: LeanDex, a semantic retrieval agent, and a Discussion Partner (a single-call LLM module). Evaluated on the Putnam 2025 dataset, the framework successfully solved 12 problems, achieving performance parity with state-of-the-art closed-source models. Furthermore, the study demonstrates that when working in synergy with human mathematicians, the agent is capable of formalizing mathematical research.

**Compliance With Llm Reviewing Policy:**

Affirmed.

**Key Questions For Authors:**

Regarding Weakness 2 (LeanDex): Do the authors plan to open-source LeanDex? If so, it is essential to provide further technical implementation details within the manuscript. If there are no plans to release it, the authors should consider revising the title or description to avoid using the term "Open" to describe the system.

Regarding Weakness 3 (Cost Calculation): Are the LLM calls made within the MCP server (e.g., those for the informal prover and discussion partner) included in the total cost/budget calculations? The authors are encouraged to provide a more detailed clarification of their cost-control strategies and the breakdown of expenditures across the different system components.

**Limitations:**

yes

**Strengths And Weaknesses:**

**Strengths**

1.  **Innovative Paradigm, Exceptional Performance, and High Potential:** Utilizing a general-purpose coding agent to solve ATP tasks is a highly promising direction. Since general programming languages and ITP languages both belong to the category of formal languages, the logic behind this paradigm shift is particularly sound. The results presented in Tables 1-3 demonstrate the significant potential of this approach.

2.  **Comprehensive Ablation Studies Providing Community Guidance:** The paper offers extensive ablation experiments across various tools and models, providing valuable guidance for model selection within the research community. Tables 4 and 5 clearly illustrate the impact of individual components on the agent’s overall performance. Furthermore, the authors are commendably transparent in stating that the agent's current configuration achieves its optimal performance specifically with Claude 4.5 Opus.

3.  **Practical Insights into Human-AI Collaboration:** The discussion in Section 5.2 regarding the verification-driven "plan–advance–review" cycle provides a framework for human-machine interaction. This contribution is instrumental in helping the research community adopt such agent-based paradigms at scale, moving beyond small-scale laboratory benchmarks toward real-world mathematical applications.

**Weaknesses**

1.  **Unclear Contribution and Potential Overclaiming:** The authors introduce "Numina-Lean-Agent". However, the majority of its MCP components appear to be derived from existing sources: the **Lean-LSP-MCP** is maintained by the community, the **informal prover** is based on prior work, and the **Discussion Partner** consists of a single LLM call. Packaging these disparate existing tools together under a new proprietary name may be perceived as an overstatement of the paper's original contribution.

2.  **Lack of Transparency in LeanDex Implementation:** The description of **LeanDex**—the only novel semantic retrieval component within Numina-Lean-MCP—is notably brief. Around line 174, it is merely described as "an agentic semantic search tool built on top of LeanExplore." Although Table 5 suggests that the absence of LeanDex does not lead to a significant drop in performance, the lack of technical detail prevents readers from understanding its specific implementation or the techniques used to enhance semantic retrieval. This lack of detail undermines the overall transparency and reproducibility of the research.

3.  **Ambiguity in Cost Calculation and System Architecture:** The authors integrated GPT-5.2 Pro, Gemini 3 Pro, and DeepSeek V3.2 as informal provers and discussion partners within the MCP server rather than within the primary agent system. This is an unconventional architectural choice. Because LLM API cost is partitioned between the Agent harness and the MCP, accurately tracking and controlling costs becomes difficult. Given that GPT-5.2 Pro is significantly more expensive than other models (168 dollars per million output tokens), the authors should clarify their cost-management strategy. This is especially pertinent given the statement in Line 209 that an approximate budget of $50 per problem was used; for a model as costly as GPT-5.2 Pro, this budget allows for only approximately 300K tokens of reasoning space, which warrants further explanation.

---

> ### Author Rebuttal · Authors · 2026-03-31
>
> Thank you for your review!
>
> **Unclear Contribution and Potential Overclaiming**
>
> We agree that the main contribution of our work is not the standalone novelty of each component, but the agentic framework that integrates and orchestrates them into a stronger formal reasoning system. Concretely, our system combines:
> - a modified version of the community-based Lean-LSP-MCP for Lean interaction
> - Leandex which we build on top of LeanExplore for theorem retrieval
> - an Informal Prover built on prior work
> - Discussion Partner, which we design as a lightweight mechanism for flexibly incorporating external LLM assistance.
>
> Built on these tools and prior work, our system shows that a general-purpose coding agent can orchestrate Lean interaction, retrieval, informal proving, and auxiliary discussion tools into a unified formal reasoning system, yielding stronger and more flexible capabilities than a single model or prover alone. To avoid any impression of overclaiming, we will revise the next version to make the role and contribution of each component clearer, and to emphasize that our primary novelty lies in the framework behind the paradigm of using a general coding agent as a formal math reasoner.
>
> **Regarding Weakness 2 (LeanDex): Do the authors plan to open-source LeanDex? If so, it is essential to provide further technical implementation details within the manuscript. If there are no plans to release it, the authors should consider revising the title or description to avoid using the term "Open" to describe the system.**
>
> We would like to clarify that LeanDex is intended to be fully open. The LeanDex API is already publicly accessible, and the codebase will be open-sourced.
>
> **Regarding Weakness 3 (Cost Calculation): Are the LLM calls made within the MCP server (e.g., those for the informal prover and discussion partner) included in the total cost/budget calculations? The authors are encouraged to provide a more detailed clarification of their cost-control strategies and the breakdown of expenditures across the different system components.**
>
> Yes, all LLM calls, including those made within the MCP server (e.g., for the informal prover and the discussion partner), are included in the total budget. While we do not currently maintain a detailed cost breakdown for each component, we provide a rough statistic for your reference: the informal prover (IP) and discussion partner (DP) together cost around $50 to run on 12 problems.

---

> > ### Author Rebuttal · Reviewer_T9xY · 2026-04-04
> >
> > I would like to thank the authors for their response. The authors have committed to revising the next version to clarify the role and contribution of each component, which effectively addresses my concerns regarding Potential Overclaiming. I also welcome the authors' statement that LeanDex is intended to be fully open-sourced.
> >
> > However, the complete absence of technical details regarding LeanDex in the manuscript still leaves me with a strong concern regarding 'Salami Slicing'. Furthermore, the authors did not provide further explanation of the methods for controlling computational costs (merely repeating the descriptions from the paper), and no relevant implementation was found in the code provided in the Supplementary Material. Therefore, my concerns in this regard remain.

---

> > > ### Author Response · Authors · 2026-04-07
> > >
> > > Thank you again for your attention and constructive feedback. We understand you still have some questions regarding LeanDex technical details and cost management, and we are happy to provide more detailed information on both aspects.
> > >
> > > **1. LeanDex Technical Details and Addressing the "Salami Slicing" Concern**
> > >
> > > First and foremost, we want to unequivocally assure you that we have no intention of "salami slicing" LeanDex into a separate publication. The initial version did not include technical details due to space constraints and the fact that Leandex is primarily built on LeanExplore, not because we want to conceal information.
> > >
> > > The primary motivation for introducing LeanDex is as follows. The previous tool, LeanExplore, was developed on Lean 4.19 and has become incompatible with the latest environments due to a lack of maintenance. Furthermore, its exclusive support for Mathlib is no longer sufficient to meet the increasingly diverse research needs, as the community is actively developing and maintaining additional repositories (e.g., FLT and other domain-specific libraries), which also require effective theorem search tools. Consequently, we are launching LeanDex based on Lean 4.26. By extending support to libraries such as FLT and PhysLean and establishing a regular update cycle, we aim to provide the community with more robust, reliable and up-to-date support.
> > >
> > > Our Leandex is an improved version based on the LeanExplore project. The specific changes are:
> > >
> > > - We rebuilt the LeanExplore database by using the then-latest versions of repositories such as Mathlib as the foundation, and updated once a month. In addition, we regenerated the informal descriptions for each definition and theorem using DeepSeek-Chat v3.1, improving their accuracy and consistency.
> > > - When the agent issues a natural language query, LeanDex does not search it directly. Instead, it first uses an LLM to refine and optimize the prompt to better capture the mathematical intent.
> > > - Because initial retrieval results can be noisy, we apply a final LLM-based re-ranking step. The LLM evaluates the retrieved candidates and sorts them to ensure that the most highly relevant definitions and theorems are returned.
> > >
> > > We will include all the details in the appendix of the new edition. By fully open-sourcing the LeanDex code (including this query-optimization and re-ranking pipeline) alongside the main agent framework, we aim to provide a complete, usable tool for the community in this single paper.
> > >
> > > We also evaluate the performance of LeanExplore and LeanDex using three queries (first two for Mathlib, third for FLT). Across all cases, LeanDex consistently demonstrates superior retrieval performance compared to LeanExplore.
> > >
> > > |Query|Theorem|LeanExplore|LeanDex|
> > > |---|---|---|---|
> > > |ChineseRemainderTheorem|Nat.chineseRemainder|×|✓|
> > > |a/b=c/diffad=bc(b,d≠0)|Int.ediv_eq_ediv_of_mul_eq_mul|×|✓|
> > > |GaloisRepresentation|GaloisRep|×|✓|
> > >
> > > **2. Cost Control Implementation and Clarification on GPT-5.2 Pro**
> > >
> > > We apologize for the lack of clarity in cost calculation, we do not have corresponding code to control cost calculation. The reason you did not find relevant implementation in the Supplementary Material is that our cost control was not handled programmatically within the code itself, but rather enforced at the API key level.
> > > Cost Control Mechanism:
> > >
> > > We managed costs by setting hard billing limits directly on the API keys. For instance, before initiating a specific task, we would allocate a dedicated API key with a strict, pre-set budget ceiling (e.g., \$ 50). If the agent exhausted this budget, the API would simply return a limit error, naturally halting the generation. We will clarify this practical setup in the manuscript.
> > > Clarification on Model Usage and Budget:
> > >
> > > We also realize we caused a misunderstanding regarding the use of GPT-5.2 Pro. To clarify:
> > >
> > > - Ablation studies only: Models like GPT-5.2 Pro and DeepSeek were used only for ablation studies on a small subset of problems to compare model capabilities. We did not calculate computational costs during the ablation experiment phase.
> > >
> > > - Main Evaluation: For our formal, full-scale evaluation of 12 problems, we exclusively used Gemini 3 Pro, which is significantly more cost-effective.
> > >
> > > - The \\$ 50 / 12 Problems Statistic: The approximate cost of \\$ 50 for the Informal Prover (IP) and Discussion Partner (DP) across 12 problems was not a theoretical calculation based on GPT-5.2 Pro's pricing. Instead, it is an empirical statistic derived directly from our actual Gemini API billing dashboard during the formal evaluation phase.
> > >
> > > We will ensure the revised manuscript explicitly distinguishes which models were used for the ablation studies versus the main evaluation, and we will clearly detail our API-key-based budget enforcement strategy.
> > >
> > > Thank you again for holding us to a high standard of rigor. We hope these clarifications and the forthcoming additions to the manuscript fully address your valid concerns.

---

### Official Review · Reviewer_UvbS · 2026-03-14

**Soundness:** 3
**Presentation:** 3
**Significance:** 3
**Originality:** 3
**Overall Recommendation:** 4
**Confidence:** 3

**Summary:**

This paper proposes Numina-Lean-Agent, a powerful formal reasoning system.

**Compliance With Llm Reviewing Policy:**

Affirmed.

**Key Questions For Authors:**

1. How much of the performance gain comes from the base model itself v.s. the tool stack and orchestration?
2. What is the marginal benefit of the discussion partner relative to the informal prover alone?

**Limitations:**

yes

**Strengths And Weaknesses:**

### Strengths

1. Solving all 12 Putnam 2025 problems is impressive.

2.  The LeanDex ablation, the informal-prover/discussion ablation, and the informal-prover model-choice ablation provide some evidence that the gains do not arise from a single source.

### Weaknesses

1. The paper’s strongest results depend on Claude Opus 4.5 plus other proprietary components, while the base-model ablation only covers a subset of tasks and shows a large drop for alternative models. The manuscript argues that other code agents could be substituted, but this is not actually demonstrated.

2. The work seems most novel as a strong systems integration effort rather than as a fundamentally new reasoning algorithm. That is acceptable, but the manuscript should be more precise about what is new relative to prior agentic formal reasoning systems such as Ax-Prover, HILBERT, Seed-Prover, and AxiomProver.

---

> ### Author Rebuttal · Authors · 2026-03-31
>
> Thank you for your review!
>
> **The core results over-rely on models like Claude Opus 4.5, with limited ablations showing sharp performance drops for alternatives. Furthermore, the claimed substitutability of other code agents lacks empirical proof.**
>
> We appreciate the constructive feedback on our framework's generalizability. Our base-model ablation was limited to a 6-problem subset purely due to computational budget constraints. To demonstrate robustness, we additionally evaluate with Gemini 3.1 Pro, which successfully solved 5/6 problems (table below). This proves our peak performance is not tied to Claude Opus 4.5. Furthermore, while open-source models like DeepSeek V3.2 yield lower performance, our framework still empowered it to solve 2/6 Putnam problems. Compared to vanilla DeepSeek's 0/6 baseline, this substantial leap demonstrates our framework's consistent value regardless of the base model's inherent ceiling. To validate code agents replaceability, we substitute Claude Code with OpenCode, a fully open-source agent. This configuration maintains competitive performance (5/6 problems solved), confirming our system does not depend on opaque framework. Collectively, these results establish our method as model and agent-agnostic. We attribute the remaining performance gap between closed and open-source models primarily to disparities in long-term planning and intrinsic agentic reasoning. Bridging this gap remains an important direction for future work.
> ||a1|a2|a3|b1|b2|b3|
> |---|---|---|---|---|---|---|
> |Claude Code+Claude Opus 4.5|V|V|V|V|V|V|
> |Claude Code+Gemini 3.1 Pro|V|V|V|X|V|V|
> |OpenCode+Claude Opus 4.5|V|V|V|X|V|V|
>
>
> **It must clearly differentiate its exact contributions from prior agentic systems.**
>
> Unlike Ax-Prover, HILBERT, and SeedProver, our framework introduces a structurally flexible approach. Ax-Prover and HILBERT rely on rigid, predetermined pipelines (e.g., fixed multi-agent cycles or strict sketch-translate-decompose workflows), while SeedProver is trained solely to invoke specific tools. These rigid designs limit scalability on complex tasks. Conversely, Numina-Lean-Agent leverages a general coding agent paradigm. Rather than following predefined loops, it autonomously determines its step-by-step workflow based on the real-time proof state. This allows seamless performance scaling by simply integrating new tools (e.g., via MCP) without architectural redesigns, establishing it as a highly extensible, general-purpose reasoning system for formal mathematics.
>
> **How much of the performance gain comes from the base model itself v.s. the tool stack and orchestration?**
>
> We appreciate this insightful question. Two empirical results demonstrate that performance gains are primarily attributable to our tool stack and orchestration. First, our ablation study (Table 5) quantifies the tools' indispensable role: on a 6-problem Putnam subset, removing LeanDex causes direct failure on B1, while ablating the informal prover and discussion partner reduces solved problems to 3/6. This confirms the tool stack actively drives proof search. Second, to intuitively gauge the base model's standalone capabilities, we evaluate Claude Opus 4.5 natively on the Putnam 2025 benchmark. The standalone model failed on all problems (0/12) under Pass@32, establishing that the base model alone is insufficient to solve these problems. In summary, while the base model's reasoning capacity is a critical prerequisite, it cannot natively generate complete, syntactically correct, and "sorry"-free formal proofs in a single pass. Our orchestration successfully bridges this execution gap, unlocking the model's latent potential to conquer all 12 problems.
>
> **What is the marginal benefit of the discussion partner relative to the informal prover alone?**
>
> As shown in table below, discussion_partner significantly outperforms gemini_informal_prover in invocations for problem A5. This stems from its superior flexibility in addressing proof details through open-ended dialogue. We also conduct an ablation study on the discussion partner using two problems (A5 and B6), and both problems become unsolvable without the discussion partner, demonstrating that this component is essential.
> |Tool|a1|a2|a3|a4|a5|a6|b1|b2|b3|b4|b5|b6|
> |---|---|---|---|---|---|---|---|---|---|---|---|---|
> |lean_diagnostic_messages|44|58|53|131|1683|103|40|59|69|121|17|212|
> |gemini_informal_prover|1|1|1|0|10|2|1|2|1|13|1|11|
> |lean_leandex|16|42|5|36|180|17|34|40|27|34|10|16|
> |lean_loogle|7|27|14|45|153|27|18|30|13|39|8|41|
> |lean_local_search|14|34|5|8|144|5|21|43|28|12|11|27|
> |discussion_partner|0|2|0|0|212|0|2|0|0|0|1|9|
> |lean_goal|10|13|21|47|714|25|7|33|13|45|10|101|
> |lean_hover_info|6|11|12|4|85|6|13|19|2|4|0|23|
> |lean_run_code|2|29|1|112|5|1|59|23|31|21|125|0|
> |lean_declaration_file|1|3|0|1|1|0|2|3|1|1|0|0|
> |lean_completions|3|0|0|1|7|1|0|1|0|2|0|0|
> |lean_multi_attempt|1|0|0|5|35|0|0|1|0|2|0|76|
> |lean_build|2|0|1|0|11|2|0|1|1|0|0|0|

---

### Decision · Program_Chairs · 2026-04-30

**Decision:**

Accept (regular)

**Comment:**

This paper introduces Numina-Lean-Agent, a novel formal theorem-proving framework that shifts away from rigid, task-specific pipelines and trained formal provers. Instead, it utilizes a general-purpose coding agent paradigm to autonomously interact with the Lean proof assistant. By integrating community tools and novel components, such as the LeanDex semantic retrieval tool, an Informal Prover, and a Discussion Partner, through the Model Context Protocol (MCP), the framework allows the agent to dynamically determine its workflow based on the real-time proof state.

Reviewers are overall positive of the paper findings and claim formulations.